# Enhanced Hybrid Ant Colony Optimization for Machining Line Balancing Problem with Compound and Complex Constraints

**Junyi Hu [1], Zeqiang Zhang [2,\*], Haixuan Qiu [3], Junbo Zhao [1] and Xuechen Xu [1]**

[1]   Rail Transit Department, Zhejiang Institute of Communications, Hangzhou 311112, China;
      hujunyi_jt@zjvtit.edu.cn (J.H.); zhaojunbo@zjvtit.edu.cn (J.Z.); 1062329939xxc@gmail.com (X.X.)
[2]   School of Mechanical Engineering, Southwest Jiaotong University, Chengdu 610031, China
[3]   Civil Aviation Department, Zhejiang Institute of Communications, Hangzhou 311112, China;
      qhx@zjvtit.edu.cn
[\*]   Correspondence: zhangzq@home.swjtu.edu.cn

**Abstract:** Targeted at the machining production line balancing problem, based on the precedence constraint relation of the present machining task, this article suggests adding practical constraints such as advanced station preparations, post-auxiliary tasks, and tool changing. The study introduced 'tight' and 'or' constraints to bring the problem definition closer to the actual situation. For this problem, a mixed-integer programming model was constructed in this study. The model redefines the machining and auxiliary processing tasks and adds new time constraints to the station. The model considers two optimisation objectives: the number of stations and the machining line balancing rate. In view of the complexity of the problem, heuristic task set filtering mechanisms were designed and added to the ant colony optimisation, to satisfy the above compound and complex constraints. The processing task chain was constructed using the rules of ant colony pheromone accumulation and a random search mechanism. The study designed a Gantt chart generation module to improve the usability and visibility of the program. Ultimately, through an actual case study of a complex box part including 73 processing elements and realising the design and planning of machining production lines that meet complex constraints by substituting algorithms, the balance rates of several groups of optimisation schemes were higher than 90%, which showed that the algorithm is effective and has a good economy and practicability.

**Keywords:** directed graph; machining technology; ant colony optimization; line balancing problem

## 1. Introduction

In the mass production customization stage of manufacturing products, the preparation process of process technology and equipment occupies a key position, which will directly affect the effective play of productivity [1]. Therefore, processing technology design is a key step in the machinery manufacturing industry, and its quality of processing technology design directly affects product quality and processing efficiency. Process design is limited by the manufacturing resources and abstract conditions in the product's own design; therefore, it is also a systematic project. Hence, Xu et al. [2] considered computer-aided process planning (CAPP) to be a link between computer-aided design (CAD) and computer-aided manufacturing (CAM), and it is also an important part of the current computer-integrated manufacturing systems. However, the product modularization and intensive design concept of today's products lead to the diversification of the product's own processing elements and constraints, and the demand for production equipment and processing conditions also presents a trend of diversification. Thus, CAPP is a very complex project [3]. Therefore, Luki et al. [4] believe that CAPP is the weakest link in today's computer-integrated manufacturing system, and it is also an area that needs further study.

As mentioned by Denkena et al. [5], the detailed steps of the process planning process include receiving and analysing design drawings, selecting raw materials of suitable shapes,

selecting the process technology, deciding the process sequence and flow, and preparing processing plans. It concludes equipment requirements, flow sequence of products in the equipment, cutters and tools, processing step design, processing parameter selection, measuring tools and tooling design, tool path planning, process cost, and time analysis. Finally, obtaining an executable CNC program, process cards, operating procedures, and other process documents. Therefore, the goal of CAPP is to transform the product from the state of design drawings into the final physical product, with reasonable efficiency and quality.

Among the above-mentioned process activities, sorting the processing tasks is the core that determines the quality and efficiency of product processing. It is also the most complex and difficult task in the entire CAPP process, which meets all the constraints and enables the smooth processing of the product. This work can be divided into three steps: (1) According to the design drawing of the product, analyse all its processing tasks and obtain the required processing equipment, tools, processing orientation, processing accuracy, and other information according to the processing tasks. (2) Analyse the mutual constraints between all processing tasks of the product, sort the processing tasks to form an effective processing task chain, and optimise task sorting to make the overall processing time the shortest or the processing efficiency the best. (3) On the basis of forming an effective processing task chain, the tasks of the processing task chain are divided into workstations in order, such that the processing time between the workstations is as balanced as possible. Step three is the machining production line balancing problem, which is a secondary optimisation based on the ordering of processing tasks by step two, and the tasks are equally distributed in each workstation. This process is carried out so that the products can be produced in the minimum cycle time between the workstations, productivity can be improved, and the idle time of production equipment can be reduced, thereby improving equipment utilisation. Therefore, the design of the machining line in the mass flow production mode has to focus on improving line balance while ensuring the machining line meets diverse machining process constraints and complicated process requirements in terms of part dimensions. Consequently, the machining line balancing problem represents a crucial research direction regarding the manufacturing system in that of its significance to the improvement of productivity and product quality.

The solution of the second and third steps of the problem has been considered NP-hard by many scholars, such as Duo et al. [6], Falih and Shammari [7], Petrovi'c et al. [8], Gao et al. [9], Huang et al. [10], and Dolgui [11]. At the same time, bionics and heuristic algorithms have been proven to solve such problems well, and they can still consider the solution efficiency and quality when the problem scale increases exponentially. Many scholars have conducted research on the sorting of processing tasks mentioned in Step two, and good results have been achieved. It is further challenging to solve the machining production line balancing problem, the number of scholars who have studied it being relatively few.

## 2. Related Work

### 2.1. Related Work for Operation Sequencing Problem in the Recent Five Years

Petrovic et al. [9] used an AND/OR network to define the flexible machining task planning and sequencing problem, taking the shortest machining time of the overall machining chain as the optimisation goal, and using the particle swarm algorithm to solve it. Hu et al. [12] introduced clustering constraints to bring the problem closer to the actual situation and proposed an ant colony optimisation to solve it. Huang et al. [10] proposed a hybrid genetic algorithm combined with a simulated annealing mechanism to solve this problem. To ensure the feasibility of the solution, sequence constraints between the tasks were incorporated into the algorithm. Su et al. [13] proposed a mixed-integer programming model and integrated an improved selection crossover strategy into a genetic algorithm to ensure the feasibility of the solution obtained after the genetic operation, thus improving the efficiency of the genetic algorithm. Dou et al. [6] proposed a discrete particle swarm optimisation algorithm, based on generating feasible task sequences, and used a

crossover mechanism to expand the feasible solution search interval. Multiple mutation mechanisms are used to increase the diversity of feasible solutions, and adaptive mutation probability is used to increase the detection ability. Liu et al. [14] introduced 'or' type network nodes, redefined the mixed-integer programming model for this problem, and solved it using a mixed evolutionary algorithm based on a genetic algorithm and simulated annealing algorithm. An adaptive evolutionary screening mechanism was used to prevent the algorithm from falling into the local optima. Gao et al. [9] proposed an intelligent water drop algorithm to solve the conventional process-chain planning problem. Falih and Shammari [7] proposed a genetic algorithm based on feasible sequence rules, which limited the search feasible region for the genetic algorithm through feasible sequence rules, thereby improving the search efficiency. The detection ability of the algorithm was improved by the hybrid crossover operator to prevent it from falling into a local optimum.

Through the above research, over the past five years, it was found that the current task sequencing problem for machining task chains is limited to conventional task priority constraints, machining orientation constraints, machining equipment, and tool constraints. However, the inspection of semi-finished products in each clamping process, the cleaning of processing tooling, the consideration of workpiece clamping time, the deburring of semi-finished products after each machine tool process, and the routine size inspection process are not considered. These are necessary tasks that occur at all times on the daily machining production line and are indispensable auxiliary tasks for processing. Neglecting these generally necessary auxiliary tasks will limit the application of optimisation results.

### 2.2. Related Work for Machining Line Balancing Problem

At present, flexible production lines based on advanced manufacturing equipment, such as machining centres, turning–milling combinations, and multi-axis motion, can be competent for the processing technology of most complex mechanical parts. However, on a production line consisting of multiple flexible machines and equipment, time-consuming bottlenecks often occur. The machine or equipment that undertakes the bottleneck process determines the rhythm of the entire production line and affects the production capacity of advanced flexible production lines [15]. Balancing the processing tasks assigned to parts among multiple machines and equipment such that the number of processing tasks allocated to each machine or equipment is similar eliminates the bottleneck process, which has always been one of the challenges faced by the manufacturing industry. In this study, a typical complex box part containing 73 machining tasks was used to evaluate the process–balance distribution of the machining line.

Production lines are designed for mass production, the assembly, disassembly, or processing of products. Manufacturing production lines can be divided into assembly and disassembly, machining, and petrochemical metallurgical process lines. Many studies have been conducted on the assembly line/disassembly line balancing problem, and intelligent optimisation algorithms have been applied to the assembly line and disassembly line balancing problem, achieving good results [16–19].

The optimisation goal of the machining line balancing problem (TLBP) is to balance the processing time of each station in order to reduce equipment idle time and minimise the production line cycle time [20], the number of stations [21,22], and the production line cost [10]. Most studies on TLBPs have focused on the assumption that parts are machined by single or multi-spindle machines [11,23,24]. Dolgui et al. [23] proposed the distribution of the machining tasks of parts to different workstations in a subset. Essafi, Delorme, and Dolgui [25,26] studied the procedure to distribute all manufacturing process tasks of a part equally among various workstations and to reduce the total number of processing equipment. Osman et al. [27] considered the processing line balance objective as the minimisation of non-productive time and proposed ant colony optimisation to solve it. Borisovsky et al. [28] sought to minimise the cost of the TLBP within the feasible solution constraints. Zhang et al. [29] regarded the part clamping and positioning scheme as the clustering constraint of the machining elements, considered the machining auxiliary

elements such as tool change and machining direction, and used ant colony optimisation to optimise and improve. Li et al. [30] proposed an improved genetic algorithm for the production line balance problem of mixed-flow parts in part families. Liu et al. [31] used an improved genetic algorithm to study the multi-objective optimisation balance problem of part machining lines. Pavel Borisovsky et al. [32] proposed exclusion and inclusion constraints for the uncertainty TLBP and used a Heuristic method to solve this problem. Cong He et al. [33] also studied the uncertainty TLBP, but the 'and' and 'or' constraints are not considered. Pavel Borisovsky [34] studied this problem with parallel machines and a simple but effective genetic algorithm is proposed for solving this problem. Kłosowski et al. [35] studied this problem by the criteria of minimisation of machining times and costs and developed and compared two models of analogous manufacturing systems. It can be seen from the above literature research that the current research trend is to consider more complex constraints or a probability model based on the random processing time of tasks.

Existing methods use positioning and clamping as the clustering criterion and the machining constraint matrix as the machining task sequence constraint, and use ant colony optimisation or a genetic algorithm to solve the machining line balance problem. However, from the perspective of setting constraints, these two constraints cannot be considered comprehensively; only auxiliary elements such as tool change and machining direction are considered, and the addition of 'or' constraints is not considered to improve the task constraint matrix [14]. Moreover, the auxiliary processing time of each station (such as incoming inspection, deburring, tool cleaning, and dimensional inspection) is not considered. Therefore, the application of the optimisation results is limited.

This paper also considered most of the constraints reported in the existing literature, such as task machining priority, tool change, machining orientation, tightness, and 'or' constraints. At the same time, additional constraints were added to each process, such as incoming material inspection, the deburring of semi-finished products, the manual positional dimensional inspection using a position gauge, and the routine inspection of dimensions. The contribution of this study is to consider most of the complex constraints and influencing factors mentioned above and propose a mathematical model that is closer to the actual production situation. Thus, a new branch has been opened up on the basis of the original research. But fundamentally, this problem also belongs to the NP-type problem in combinatorial optimization, and compared with the existing machining line balance problem, it contains more constraints. To address various complex constraints, ant colony optimisation was proposed based on a feasible task selection strategy. Finally, a complex box product was used as an example to verify the practicability of the algorithm.

## 3. Definition of Process Constraints for the Balancing Machining Line Problem

The balancing problem of machining production line has many similarities with the balancing problem of unilateral assembly line. The difference lies in the consideration of special properties that need to be added for more tasks. For example, tools, tooling, processing orientation, machine type, tool changing process, auxiliary processes, etc., are all factors that cannot be ignored for processing tasks. Therefore, when studying this problem, it is necessary to first briefly describe the problem through a small-scale case, so as to facilitate the subsequent discussion.

### 3.1. Simple Bracket Product Processing Case of Machining Production Line (TLBP) with Advanced Station Preparation and Post-Auxiliary Tasks

To facilitate the understanding of the additional constraints added in this study, a processing case of a simple part was used for analysis and description. Figure 1 shows a simple part with eight machining elements. There are three datum elements: machining element 1 is a plane (datum A), machining element 2 is a plane (datum B), and machining element 3 is a hole (datum C). Machining elements 4 and 5 were placed on two sides.

Processing element 6 is a blind hole located on processing element 5. Processing element 7 is a step located on processing element 5, and processing element 8 is a through hole.

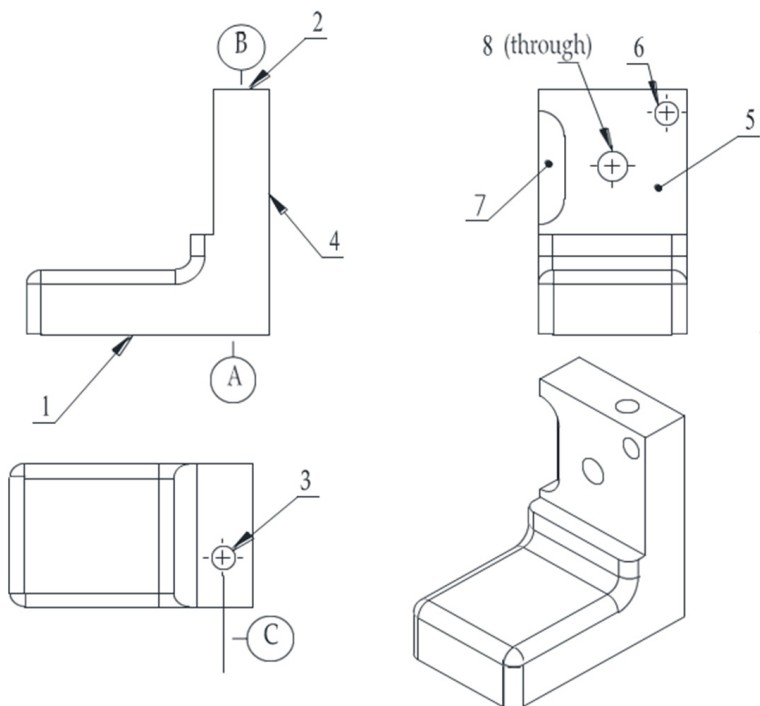

**Figure 1.** Simple bracket product processing case.

Figure 2 shows a directed graph of the sequence constraints between all machining tasks of the part shown in Figure 1. The sequence constraints between two tasks use solid arrows, and the 'or' type constraints between two tasks use dashed arrows with tight constraints indicated by a dashed box. For example, tasks 1 and 2 are normal sequence relationship constraints. Tasks 8 and 4 as well as 8 and 5 are 'or' type constraints. Tasks 6 and 8 are tight constraints. The tasks with tight constraints should be placed in the same station. The tasks with 'or' type constraints can be processed when either of its immediate priority task has been processed. The tasks with normal precedence constraints can be processed when all of its immediate priority task has been processed. The meanings of all the variables in the Figure 2 are listed in Table 1. All of the tasks in Figure 2 should be processed by the machining line.

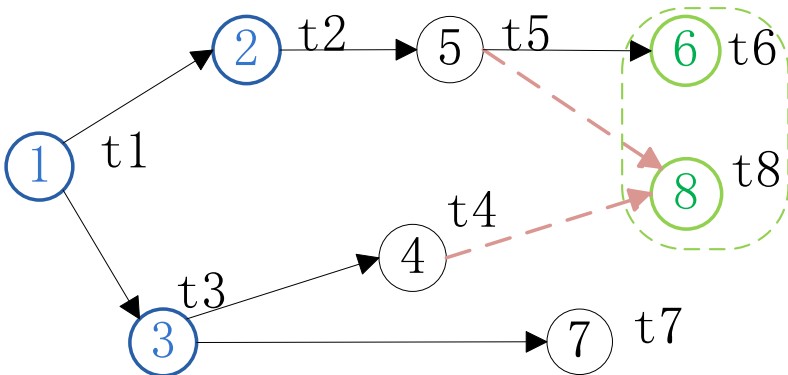

**Figure 2.** Diagram showing the processing constraint matrix.

Table 1 lists the machining content, machining time, tool, and equipment type for all machining tasks. The machining operation direction of the machining task and the surface or datum to which the machining task belongs are listed in this table.

Figure 3 shows a simple schematic of the processing production line for a part. The production line consisted of five stations, each of which contained multiple processing tasks. After the parts to be processed are positioned and clamped on the machine tool of the station, the processing tasks of all processing components in the station are completed, and the parts are then transferred to the subsequent stations until all the processing contents are completed. All machining tasks at the same station should have the same machining direction constraints on the part. The sum of the working times of all tasks at the same workstation should be less than the cycle time. In the figure, $p_z$ and $p_d$ represent the first and last auxiliary processing tasks of each non-end station, respectively, and $p_f$ represents the tail auxiliary processing tasks of the last station.

**Table 1.** Task operation procedure, time, operation direction, and tool requirement for the simple bracket part.

| Task Number | Processing Content | Processing Times (s) | Plane It Belongs to | Equipment Required | Cutting Tool Type | Machining Direction of Parts |
|---|---|---|---|---|---|---|
| 1 | Datum A | $t1$ | Datum A | Machining centre | φ30 Milling cutter | Bottom surface |
| 2 | Datum B | $t2$ | Datum B | Machining centre | φ10 combined drilling & reaming | Top surface |
| 3 | Hole φ5 (Datum C) | $t3$ | Datum C | Machining centre | φ5 combined drilling & reaming | Top surface |
| 4 | plan 4 | $t4$ | plan 4 | Machining centre | φ30 Milling cutter | Right surface |
| 5 | plan 5 | $t5$ | plan 5 | Machining centre | φ30 Milling cutter | Left surface |
| 6 | The hole φ6 on plan 5 | $t6$ | plan 5 | Machining centre | φ6 drill | Left surface |
| 7 | plan 7 on plan A | $t7$ | plan 5 | Machining centre | φ20 Milling cutter | Left surface |
| 8 | Hole φ12 | $t8$ | plan 5 | Machining centre | φ12 drill | Left surface |

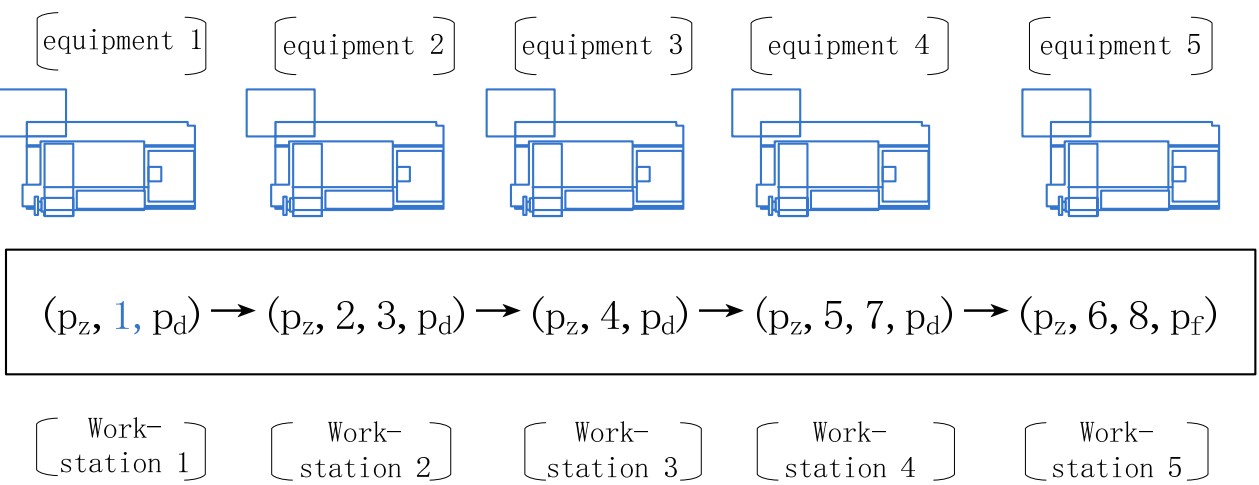

**Figure 3.** Process scheduling assignment scheme satisfying the constraints of Figure 2.

Figure 4 shows a Gantt chart that satisfies the task-scheduling scheme shown in Figures 2 and 3. Because tasks 2 and 3 use different machining tools, a tool change time must be inserted between the two tasks that also occur in tasks 6 and 8. In addition, the operation preparation time at the beginning of each station was set, and the necessary auxiliary time, such as chip removal and inspection, was added after all tasks of each station were completed. To simplify the problem, this study made the assumption that the preparation time (set-up time) is equal at each machine for each operation. It suggests that operations with times $t1$, $t2$, $t4$, $t5$, and $t6$ are realized in the same start time.

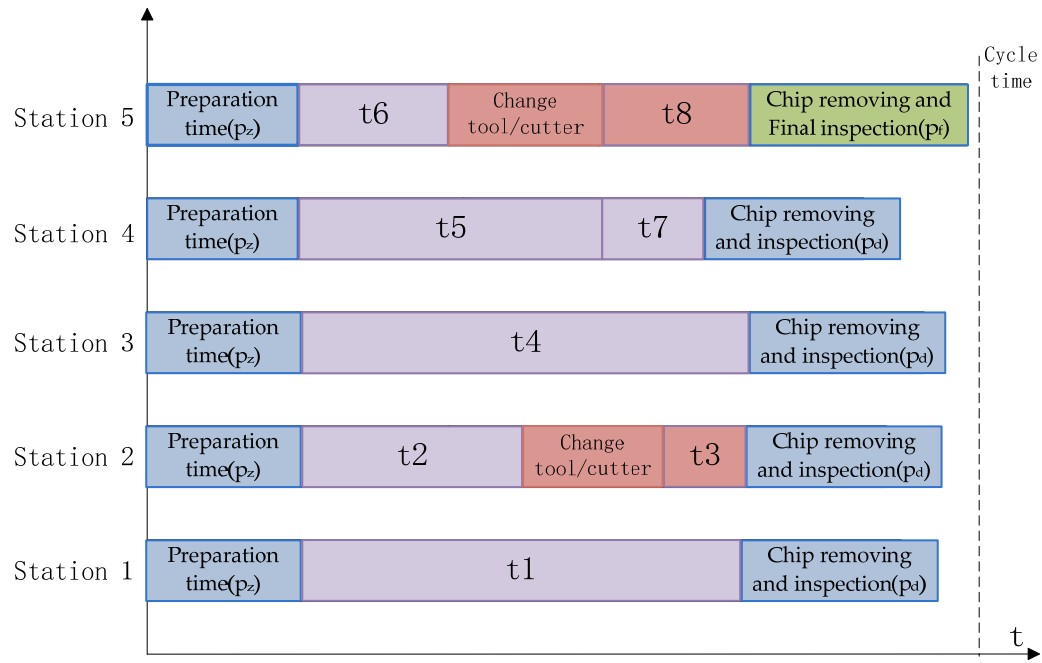

**Figure 4.** Gantt diagram of 5-station process layout satisfying the constraints of Figures 2 and 3.

*3.2. Mathematical Model for Balancing Problem of Machining Production Line (TLBP) with Advanced Station Preparation and Post-Auxiliary Tasks*

In the machining process, multiple steps such as roughing, semi-finishing, and finishing are often required to complete the processing of one size. To simplify the difficulty in solving the problem, the continuous processing steps of a unit of a certain size can be regarded as integrated into a working step. At the same time, this assumption is also consistent with the actual work step combination, such as drilling and reaming combination [36], drilling and boring combination, drilling and tapping combination, and rough and fine milling surface combination. This assumption is also in line with the current situation of the wide application of composite tools such as composite taps and drill reamers. On this basis, the process sequence constraint relationship matrix [31] between each processing element can be proposed as

$$A = \begin{bmatrix} a_{11} & a_{12} & \cdots & a_{1n} \\ a_{21} & a_{22} & \cdots & a_{2n} \\ \vdots & \vdots & \ddots & \vdots \\ a_{n1} & a_{n2} & \cdots & a_{nn} \end{bmatrix} \tag{1}$$

If dimension element $j$ is allowed to be processed only after dimension element $i$ is processed, then $a_{ij} = 1$; if the dimension element $j$ wants to be allowed to be processed, it only needs to complete the processing of either dimension element $i$ or dimension element $k$, then $a_{ij} = a_{kj} = 2$, that is called 'or' type constraint [14]; if either $i$ or $k$ is processed, the sequence constraint relationship between the other factor and $j$ becomes invalid and can be set to $-1$; if dimension element $j$ must be processed immediately after the processing of dimension element $i$ is completed, then $a_{ij} = -2$, that is, $i$ and $j$ are tight constraints; and if there is no sequential constraint relation between dimension element $i$ and dimension element $j$, then $a_{ij} = 0$; In summary, the possible values of $a_{ij}$ are as follows:

If processing task $j$ is only allowed to be arranged after the processing of task $i$ is completed, then $a_{ij} = 1$. If the premise of allowing the processing of task $j$ is only if either task $i$ or $k$ is processed first, then $a_{ij} = a_{kj} = 2$, which is called 'or' type constraint [14]; when any one of the tasks $i$ or $k$ has completed the processing, the sequence constraint relationship between the other remaining task and task $j$ is invalid, which can be set to $-1$;

if task $j$ must be processed immediately after task $i$ is processed, then $a_{ij} = -2$, that is, $i$ and $j$ are closely constrained; and if there is no sequence constraint between task $i$ and task $j$, $a_{ij} = 0$. In summary, the possible values of $a_{ij}$ are as follows:

$$a_{ij} = (-2, -1, 0, 1, 2) \tag{2}$$

Compared with literature [2–18], the addition of 'or' type constraints improves the order constraint relation matrix and provides more possibilities for task scheduling. Compared to the literature [14], tight constraints were added. When arranging a machining surface and two positioning pins or other multi-positioning datum elements, they are set as tight constraints, such that the machining tasks with the function of evaluation criteria are preferentially arranged in the same station, and the machining accuracy of the evaluation criteria is improved. The processing tasks and evaluation benchmarks that require high mutual positional accuracy can also be arranged in the same station to ensure that the dimensional accuracy requirements are satisfied. The processing positioning benchmark and dimension evaluation benchmark element take precedence over the processing task arrangement method of the dimension element being evaluated and are more in line with the process arrangement principle. Simultaneously, auxiliary tasks such as pre-treatment and deburring treatment processes, as well as size and appearance of inspection tasks, are added to each station, which is more in line with the actual situation of the machining production line.

The mathematical model of the production line balancing problem (TLBP) is more complex than that of the unilateral assembly line balancing problem (ALBP) [37–39]. In addition to the task priority relationship constraints and cycle time constraints similar to the unilateral production line balancing problem, the mathematical model for TLBP should also meet the requirement constraints of processing equipment types, tool type demand constraints, and processing direction constraints; thus, the constraints are more complex. At the same time, this study $\in$ also considered the preliminary preparation tasks, later auxiliary tasks, and tool change time of each station:

$$S_x \cap S_y = \{p_z, p_d\} \tag{3}$$

$$S_x \cap S_m = S_y \cap S_m = \{p_z\} \tag{4}$$

$$Q_x \cap Q_y = \Phi \tag{5}$$

$$\text{numel}(Q_k) = \text{numel}(S_k) - 2 \tag{6}$$

$$\sum_{k=1}^{m} \text{numel}(Q_k) = n \tag{7}$$

$$T_k \leq C, \ T_k = \sum_{i \in S_k} t_i + u_k \times p_{ht} + \left(1 - e^{k-m}\right) \times p_{dt} + e^{k-m} \times p_{ft} + p_{zt} \tag{8}$$

$$\forall i, j \in Q_k, o_i = o_j; \tag{9}$$

$$\forall i, j \in Q_k, E_i = E_j; \tag{10}$$

$$u_k = \sum_{v < \text{numel}(Q_k)} \left\lceil \left| 1 - e^{\left| g_{q_{v,k}} - g_{q_{v+1,k}} \right|} \right|^{1/M} \right\rceil \tag{11}$$

$$\forall \, i \in Q_k, j \in Q_z \, \text{if} \, a_{ij} = 1, \text{then} \, k \leq z \tag{12}$$

$$\forall \, i \in Q_k, j \in Q_z \, \text{if} \, a_{ij} = 2, \text{then} \, k \leq z \tag{13}$$

$$\forall \, i \in Q_k, j \in Q_z \, \text{if} \, a_{ij} = -2, \text{then} \, k \leq z \tag{14}$$

$$Q_k = \{i | i \in Q_k\}, S_k = Q_k \cup \left\{ p_{zt}, \left(1 - \lfloor e^{k-m} \rfloor\right) \times p_{dt}, \lfloor e^{k-m} \rfloor \times p_{ft} \right\} \tag{15}$$

$$\Omega = \{1, 2 \ldots n\} \tag{16}$$

$$LB = \left( \sum_{k \leq m} T_k - \sum_{k \leq m} (u_k \times p_{ht}) \right) \div (C \times m) \tag{17}$$

$x, y, k, z = 1, 2 \ldots, m; x < y < m$

$S_x, S_y, S_k, S_z$: = Station $x$, Station $y$, Station $k$, and Station $z$

$\Phi$: Null set

$n$: Number of tasks

$m$: Number of stations

M: A very big integer

$u$: The number of times the cutters and tools were switched

$p_h$: Cutter/tool change procedure

$p_z$: Advanced station preparation task

$p_d$: Post-auxiliary tasks, such as inspection and deburring

$p_f$: Post-auxiliary tasks, such as final inspection and deburring

$p_{ht}$: Average cutter/tool change time

$p_{zt}$: Average preparation time

$p_{dt}$: Average chip removal and inspection time for stations 1 to $m - 1$

$p_{ft}$: Average chip removal and final inspection time for final station $m$

$Q_k$: Machining tasks at station $k$

numel($Q_k$): number of machining tasks at station $k$

numel($S_k$): number of all tasks at station $k$

$u_k$: Number of cutter/tool changes at station $k$

$q_{v,k}$: Machining task at workstation $k$ with sequence number $v$

$v$: Sequence number for the machining task in workstation $k$

$E_i$: Equipment requirements for task $i$

$o_i$: Machining direction requirement for task $i$

$g_i$: Tool/cutter requirement for task $i$

$t_i$: Machining time for task $i$

$\left\lfloor e^{k-m} \right\rfloor$: The Control variable whether to turn on the final station

$LB$: Line balance rate

Equation (3) indicates that workstations $x$ and $y$ have only two identical tasks (both $x$ and $y$ are not the end workstations). Equation (4) indicates that there is only one task between workstations $x$ and $m$ and between workstations $y$ and $m$ (both $x$ and $y$ are not the last workstation, and $m$ is the last workstation). Equation (5) indicates that the processing task can only be assigned to a certain station and cannot be repeatedly assigned to multiple stations. Equation (6) indicates that each station had two auxiliary tasks. It can be seen from Equation (7) that all processing tasks should belong to a certain station and that no omission is allowed. It can be seen from Formula (8) that the cumulative processing time of each station element ($\sum_{i \in S_k} t_i$), the cumulative transformation time of tool ($u_k \times p_h$), and the sum of the preparatory and post-auxiliary task times should be less than the time *C*. It can be seen from Equation (9) that all processing tasks in the same station should have the same operation direction. Equation (10) indicates that all processing tasks at the same station should have the same equipment requirements. Equation (11) is the number of tool changes and tool changes at a certain station. Equation (12) shows the priority-order constraint in the task assignment process. Equation (13) represents an 'or' Equation (14) represents a tight constraint. Equation (15) represents the set of all tasks belonging to workstation $k$. Equation (16) represents the set of all tasks. Equation (17) indicates that the LB cannot include the cumulative transformation time of tool.

## 4. Hybrid Ant Colony Optimization for Optimization of the Machining Line Balancing Problem

### 4.1. Heuristic Task Set Filtering Module for Hybrid Ant Colony Optimization

To meet the special case of 'or' type constraints and tight constraints in the solution construction process, and, considering the increase in the total work time of changing tools and preparing work steps, the task of ant colony optimisation searching in the selectable set should satisfy all the above constraint conditions. Therefore, in this study, a temporal

constraint set generation module, a tool constraint set generation module, and a tight constraint set generation module were used as heuristic rules to satisfy multiple composite constraints. The overall algorithm flowchart of the heuristic task-set generation module is shown in Figure 5.

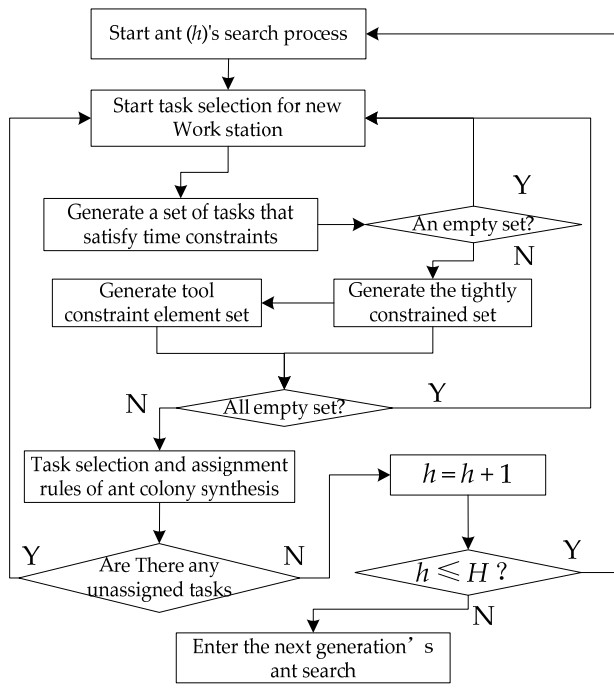

**Figure 5.** Ant colony optimization with heuristic rules.

*4.2. Introduction of Heuristic Task Set Filtering Module for Hybrid Ant Colony Optimization*

When the element sequence is formed by ant colony optimisation, the candidate element set is generated in step *s*, and then the elements are selected by the heuristic ant colony hybrid search rule until the selection of *n* elements is completed, forming a complete element processing chain.

(1) Time constraint rules: comprehensively consider 'or' constraint, as well as conventional precedence relationship constraint and cycle time constraint. The process of determining whether a certain element *j* can be added to the initial set of optional elements is as follows:

    (1.1) Set the current workstation to *k*, set $Qj = \{i \mid a_{ij} = 1 \text{ or } a_{ij} = 2 \text{ or } a_{ij} = -2\}$ and establish $Qj \in \cup_{h=1}^{k} S_h$. This indicates that all the pre-ordering tasks of element *j* have been assigned, and then proceed to the next step as follows.

    (1.2) Assume that the scheduled task completion time of the current workstation *k* is $t_{\text{temp}}$. Then, the following candidate task set $D_s$ can be preliminarily screened according to the beat time *C* constraint by $D_s = \{j \mid t_{\text{temp}} + t_j \leq C\}$.

(2) Tool constraint rules: Consider the difference between the turret tool of the current workstation *k* and the tool used by task *j* to be assigned to consider the tool change time and further filter the time constraint set $D_s$ in (1):

    (2.1) If *j* is the first task of the current workstation, the station preparation time $p_z$ should be considered.

    (2.2) If *j* is not the first task of the current workstation, check whether the tool used in task *j* is the same as the tool at the end of the current workstation; otherwise, consider the tool change time.

    (2.3) Based on the above conditions, the set T*s* conforming to the tool constraints is obtained.

(3) Based on (1) and (2), compact constraint conditions are further considered to generate the compact constraint set $j_s$, and the heuristic ant colony search rule is used to select the processing elements.

Through this heuristic task set filtering module, a feasible set of optional tasks can be provided when the task processing sequence is constructed, and then the ant colony algorithm is used to randomly select tasks to form a feasible solution. Similarly, this heuristic task set filtering module can also be combined with the standard particle swarm algorithm used in [40] to solve the research problem of this paper. Therefore, it can be used as a comparison of the enhanced hybrid ant colony algorithm in this paper.

### 4.3. Heuristic Ant Colony Search Mechanism

The ant colony optimization (ACO) algorithm is abstracted from the communication behaviours of ants, which are communicated by pheromones, to determine the best path from the starting point to the destination [37,41,42]. The ant colony optimisation used in this problem selects the task from the task set in each step, when the task set is established by the heuristic rules introduced in Section 4.1.

To satisfy the tightness constraint, it is introduced into the ant colony search rule. Therefore, a hybrid heuristic search mechanism was formed by combining tightly constrained priority, random search, and heuristic task information. The mechanism is as follows:

$$s = \begin{cases} \quad I_1 : \text{tight constraint priority } j \in J_s \\ I_2 : \ P_{js} = \dfrac{\left(\sum_{h=1}^{s} \tau_{jh}\right)^{\alpha} (w_j)^{\beta}}{\sum_{j \in T_s} \left(\sum_{h=1}^{s} \tau_{jh}\right)^{\alpha} (w_j)^{\beta}} \ 0 \leq r \leq r_1 \\ I_3 : \ \text{radom search } j \in T_s \ r_1 < r \leq 1 \end{cases} \tag{18}$$

$s$: task sequence number

$P_{js}$: the probability that task $j$ will be selected at the position of task sequence $s$

$r$: random number between $(0, 1)$

$r_1$: between $(0, 1)$, rule switchover threshold from $I_2$ to $I_3$

$\alpha, \beta$: parameters that determine pheromone intensity and the relative importance of heuristic information

$\tau_{jhhhhhh}$: pheromone of task $j$ at sequence place $h$

$\sum_{h=1}^{s} \tau_{jh}$: cumulative value of information elements of element $j$ at sequence positions from 1 to $s$

$w_j$: heuristic information of task $j$, $w_j = 1/t_j$, the ant colony search will be affected by the processing time of tasks, and tasks with short processing times will more likely be selected.

The improved ant colony optimisation integrates $I_1$, $I_2$, and $I_3$ rules, which are referred to as ACOPRO in this study. The common ant colony optimisation only integrates the $I_1$ and $I_2$ rules, which are represented by ACO in the following paper. The improved ant colony optimisation is more likely to prevent the algorithm from falling into local optimisation and has a better global search ability because of the introduction of random search rules $I_3$.

### 4.4. Pseudocode for Ant Colony Optimization and Gantt Diagram Display Module

An iterative search is completed by the ant colony optimisation, and the processing task chain is constructed according to the constraints of the sequence relationship between the processing tasks, or the constraints of the type, tight constraints, and processing orientation; the formation of the task set of each station is completed. In the process of task selection, it is also necessary to simultaneously consider the tool requirements of the task, machining orientation, other practical conditions, and to integrate auxiliary machining tasks into the beginning and end of each station. The task selection process for each position of the task chain was completed using the comprehensive information search rule of the ant colony. In the iterative process, the ant colony pheromone was updated. Thereafter, through the Gantt chart display module, the information contained in the optimal solution is visualised,

which is convenient for process planners to intuitively understand the overall situation of the solution. The pseudo-code of the overall algorithm is as follows:

| No. | Details for hybrid ant colony search algorithm and Gantt chart display module |
|---|---|
| 1 | Read task sequencing constraint data from an Excel file |
| 2 | Read the information of processing position, cutting tools, and processing time of task from the Excel file |
| 3 | Enter the number of iterations $V$ and the number of ants $H$, and set the initial iterative control variable $v$ (initial value is 1) |
| 4 | Start the search process for generation $v$ and set $h$ to 1 |
| 5 | Start the search for the $h$ ant in generation $v$, set $s = 1$, $m = 0$ |
| 6 | Start a new station, $m = m + 1$. |
| 7 | Build a task set that satisfies the cycle time constraint |
| 8 | According to formula 5, filter from the set of cycle time constraint tasks to form a set that satisfies the sequential constraint relationship |
| 9 | Filter from the set of successive constraints to form a set that satisfies the constraints of the close relationship |
| 10 | Filter from the set of close relationship constraints to form a set that satisfies the tool relationship constraints |
| 11 | Enter the ant colony comprehensive task selection process |
| 12 | If the tight constraint set is not empty, go to the next step. If it is empty, go to step 17 |
| 13 | According to Formula 11, priority is given to the tasks that satisfy the tight constraints |
| 14 | According to the selection result of step 13, analyse whether this task is consistent with the current tool information of the current station, and then proceed to the next step |
| 15 | Update the accumulated processing time information of the current station, the current tool information of the current station, and then proceed to the next step |
| 16 | Update the task chain position $s = s +1$, if $s$ is less than or equal to $n$, return to step 7. If $s = n + 1$, go to step 22 |
| 17 | If the tool or time constraint set is not empty, go to the next step. If they are all empty, go to step 6 |
| 18 | According to Formula 11, generate random number $r$, and enter the next step |
| 19 | If r is less than or equal to $r_1$, select the task according to rule $I_2$. If $r$ is greater than $r_1$, select the task according to rule $I_3$, and go to the next step |
| 20 | According to the selection result in step 19, update the accumulated processing time information of the current station, the current tool information of the current station, and then proceed to the next step |
| 21 | Update the position of task chain $s = s + 1$, and if $s$ is less than $n$, return to step 7. If $s = n + 1$, go to the next step |
| 22 | $h = h + 1$, if $h$ is less than or equal to the set number of ants $H$, return to step 5; otherwise, go to the next step |
| 23 | Update the pheromone of all ants of the current generation $v$ and go to the next step |
| 24 | $v = v + 1$, if the current iteration number $v$ is less than or equal to $V$, return to step 4; otherwise, go to the next step |
| 25 | Ant colony optimization iteration is completed |
| 26 | Enter the Gantt chart display module |
| 27 | Read the optimal solution of the task scheduling sequence and the task information |
| 28 | Read the station arrangement information cell of the optimal solution, the number of stations $m$, and the cycle time $C$ |
| 29 | Enter the Gantt chart display algorithm flow, and set the initial iteration value $g = 1$ |
| 30 | Open the display of station $g$, and proceed to the next step |
| 31 | Read the station arrangement information cell of station $g$ |
| 32 | Read the task number in order d from the cell, the number of tasks in cell D, and set the task display iteration variable $d$; the initial value of $d = 1$ |
| 33 | Whether this task is the first task of the current station, if it meets the conditions, go to the next step; otherwise, go to step 35 |
| 34 | Set the task display colour to blue, display the task progress bar according to the task information, and display the start time and end time. Go to Step 37 |
| 35 | Judge whether the tool information used in this task and the previous task is the same, if the display colour is red. If not, go to the next step |
| 36 | The task progress bar is displayed based on the task information, and the start time and end time are displayed. Go to the next step |
| 37 | $d = d + 1$, if $d$ is less than or equal to D, return step 32; otherwise, go to the next step |
| 38 | $g = g + 1$, if $g$ is less than or equal to the number of stations m, return step 30; otherwise, go to the next step |
| 39 | Gantt chart display algorithm is finished, and the program is finished |

## 5. Case of Balance Arrangement of a Box Machining Process Production Line

### 5.1. Evaluation Datum Analysis of the Box Body

Figure 6 shows the key evaluation criteria for cabinets. Among them, A, D, and E were used as evaluation standards for B and G, and B and G were used as evaluation standards for C and F. The A, D, and E benchmarks are the most important evaluation benchmarks; therefore, they were arranged in the first process. To ensure mutual positional accuracy between the datum elements, D and E should be set as tight constraints (put D and E in the same station for processing, only one clamping and positioning is required; thus, the machining accuracy of the machine tool itself can guaranteed the mutual positional accuracy between D and E). Similarly, the B and G data must be machined in the same operation, setting them as tight constraints. Because the C and F benchmarks are the evaluation benchmarks for other machining tasks on the plane where they are located, the same applies to the C and F benchmarks.

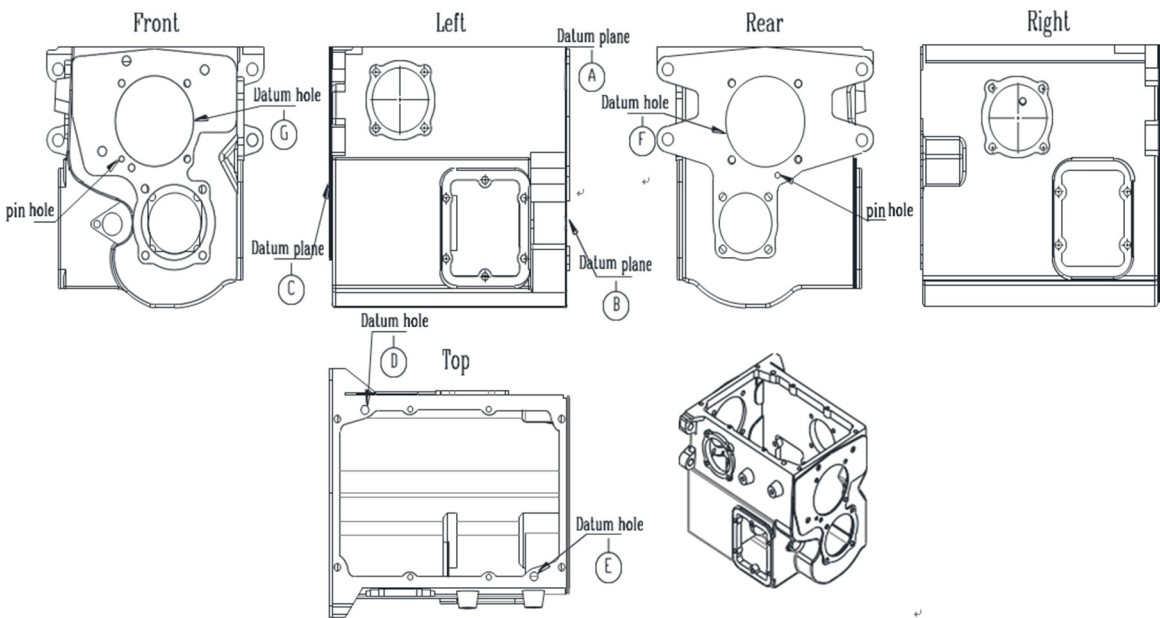

**Figure 6.** Drawings of the box key evaluation benchmark from three views.

*5.2. Analysis of Box Processing Process*

The box in Figure 6 contains 73 machining elements belonging to the top, left, and right sides, as well as the front and back of the part. Because core evaluation criteria A, D, and E are located on the top surface, the processing of A, D, and E is placed in the first station when arranging the process. The B and G datum planes were located in front of the part, and the A, D, and E datum planes were used as evaluation elements. Therefore, the A, D, and E datum planes can be used as positioning datum planes to process the B and G datum planes. Similarly, C and F data can be processed by either A, B, E, or B, G data as the positioning data.

From the above, owing to the constraints of the processing sequence between the benchmarks, the processing tasks that are grouped in the same processing orientation as the above benchmarks naturally inherit the sequence constraints between the benchmarks. All the machined hole elements located on the machined surface can be simultaneously grouped into the subsequent machining tasks of the machined surface. According to the above two principles, the 73 processed elements of this box part can be compiled into a processing sequence constraint-directed graph. Tasks 74 and 75 are deburring and inspection tasks at the end of the process; therefore, they are not included in the figure. For ease of understanding, the overall processing sequence constraint diagram is shown in Figures 7 and 8. The feature details for all 73 tasks can be found in Appendix A (Figures A1–A5) and Appendix B (Table A1).

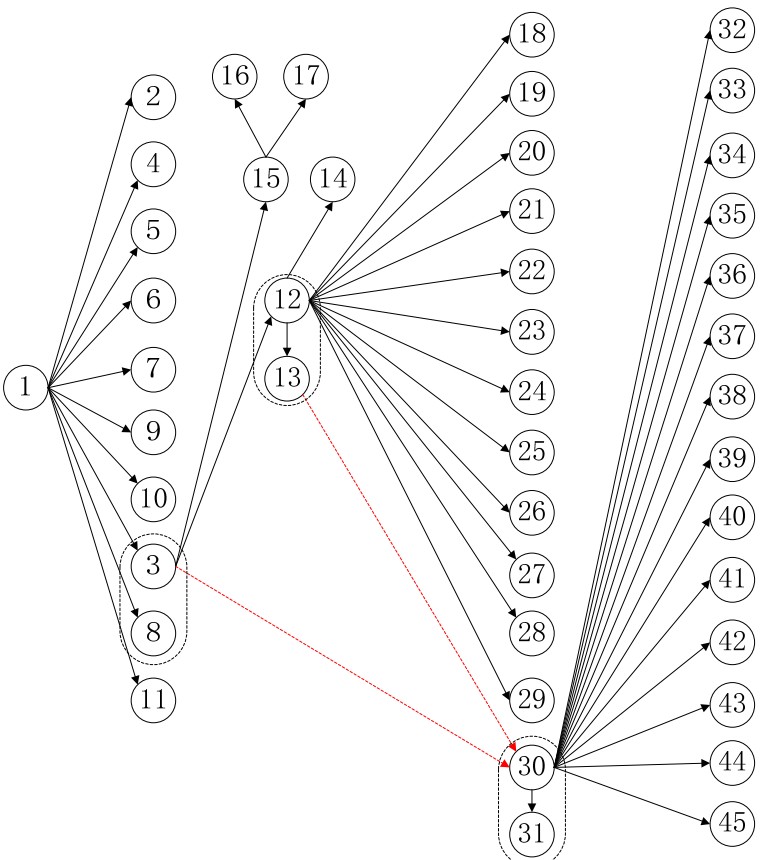

**Figure 7.** Processing constraint relationship 1.

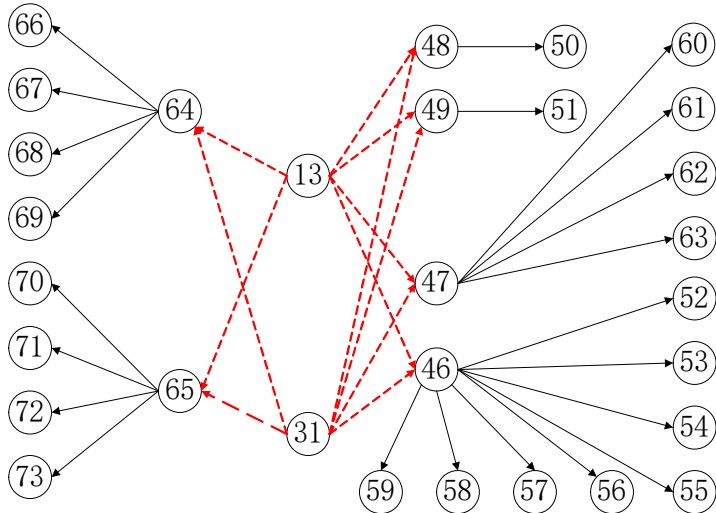

**Figure 8.** Processing constraint relationship 2.

## 6. Algorithm Verification and Instance Application

If the annual output is 13,000–15,000 units (working week: 50 weeks, 12 shifts per week, 8 h per shift), the weekly production capacity target is 260–300 sets; thus, the production capacity per shift is 22–25 sets. The cycle time (*C*) was pre-set as 19–22 min. The ant colony optimisation was used to calculate this TLBP problem. The minimum of m (the calculated number of stations, unit: piece) and the maximum of *LB* (line balance rates are the optimisation objectives).

### 6.1. Algorithm Verification by Cycle Time (1170 s)

The calculation performances of ACOPRO and ACO were compared with the pre-set cycle time of 1170 s. The common parameters of ACOPRO and ACO were set as $\alpha = 1$ and $\beta = 2$, respectively. The number of ants was set to 10, and the number of iterations was set to 20. ACOPRO's unique random search control threshold $r_1$ was set as 0.5 and 0.9, corresponding to ACOpor1 and ACOpro2, respectively. The cycle time in the above example was set to 1170 s, and ACO, ACOpro1, and ACOpro2 were substituted 20 times. The average distributions of the obtained optimisation results are shown in Figures 9 and 10.

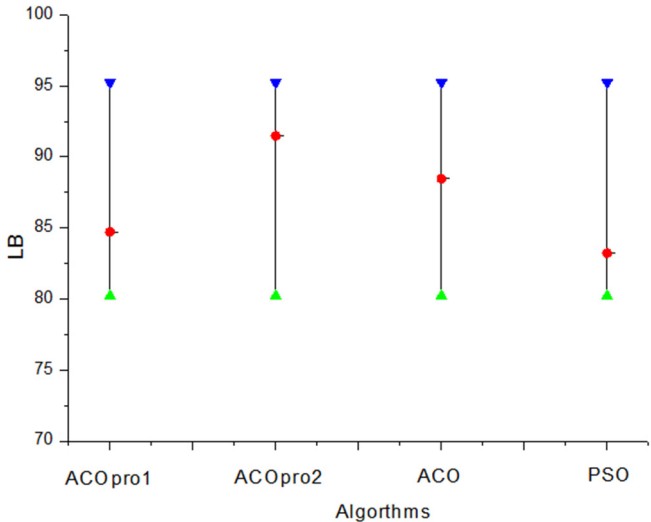

**Figure 9.** Minimum, average, and maximum values of the *LB* obtained by the comparison algorithms.

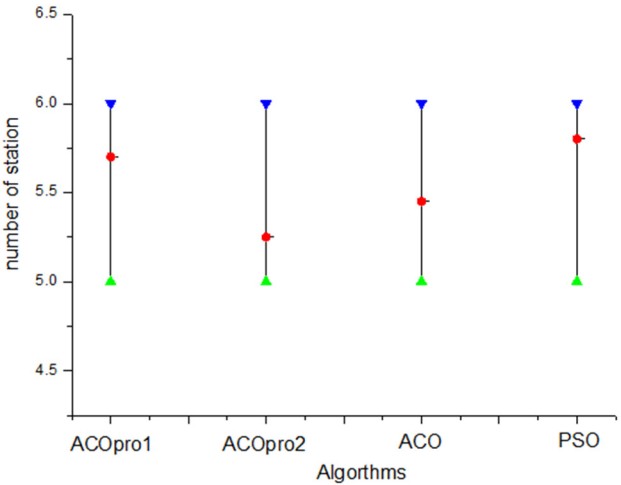

**Figure 10.** Minimum, average, and maximum values of station number obtained by the comparison algorithms.

As shown in Figures 9 and 10, when the random search control threshold $r_1$ of ACO-PRO is set to 0.5, the mean value of the optimisation results is worse than that of ACO. When $r_1$ was set to 0.9, the mean value of the optimisation results was better than that of ACO. It can be seen that excessive participation in the random search mechanism will have the opposite effect, which will disrupt the convergence trend of the ant colony optimisation itself. Therefore, ACOpro2 with an $r_1$ value of 0.9 was used as an optimisation method for a detailed analysis in the case processing example below. At the same time, ACOpro2 also

shows its superiority in comparison with the standard particle swarm algorithm with the addition of the heuristic task set filtering module in this paper on the basis of literature [40].

### 6.2. Verification with Box-Part Examples by ACO-pro2

For this TLBP problem, the pre-set cycle time was refined into an arithmetic sequence with a step difference of 10 s, and ant colony optimisation was used for the calculation. The number of ants was set to 10, and the arithmetic algebra was set to 100. The calculation results are listed in Table 2. It can be observed from Table 2 that when the cycle time was 1170 s, the balance rate of the overall processing line was the highest, reaching 95.23%. All the solutions in Table 2 can meet the production capacity requirements, but when the cycle time is shortened to 1160 s or 1150 s, the single-shift production capacity will only increase by 0.3 to 0.4 units and the balance rate of the processing line will drop by 9.58% to 32.72%, resulting in equipment and personnel resources. It is a great waste, and due to the increase in the amount of equipment, the capital investment and personnel costs are increased; hence, the rhythm of 1170 s is the preferred solution. Table 3 lists the processing start time and processing completion time of each task in each station and provides the tool change times/tool change time information of this station.

**Table 2.** Comparison of test results of each cycle time.

| No. | CT (s) | ACO-pro2 | |
| --- | --- | --- | --- |
| | | *m* | *LB* |
| 1 | 1300 | 5 | 85.71 |
| 2 | 1290 | 5 | 86.37 |
| 3 | 1280 | 5 | 87.05 |
| 4 | 1270 | 5 | 87.73 |
| 5 | 1260 | 5 | 88.43 |
| 6 | 1250 | 5 | 89.14 |
| 7 | 1240 | 5 | 89.85 |
| 8 | 1230 | 5 | 90.59 |
| 9 | 1220 | 5 | 91.33 |
| 10 | 1210 | 5 | 92.08 |
| 11 | 1200 | 5 | 92.85 |
| 12 | 1190 | 5 | 93.63 |
| 13 | 1180 | 5 | 94.42 |
| 14 | 1170 | 5 | 95.23 |
| 15 | 1160 | 6 | 85.65 |
| 16 | 1150 | 8 | 62.51 |

From the task allocation details shown in Tables 3 and 4, it can be observed that there are more workstations in Table 4, and the distribution of processing tasks among the workstations is increasingly uneven. In the allocation scheme with a cycle time of 1150 s, only one processing task 31 is allocated to station four, only one processing task 53 is allocated to station seven, and only one processing task 13 is allocated to station eight, which is the root cause of the low balance rate of this scheme. Because task 31 is located on the post-processing surface of the box part and the processing elements of the post-processing surface are concentrated at station three, the sum of the operation time of station three reaches 1065 s. If the processing time of task 31 is added, the cycle-time constraint will not be satisfied. Task 31 cannot be arranged to be processed on other machining surfaces; therefore, task 31 can only be located in one workstation alone. Similarly, available tasks 53 and 13 can only be assigned to a single workstation.

**Table 3.** Optimization results of ant colony optimization beat 1170 s scheme.

| No. | Time Details for Task Assignment | Tool Switching Times/Time (s) | Accumulated Time |
|---|---|---|---|
| 1 | 74(0, 30), 1(30, 987), 11(987, 994), 10(994, 998), 5(998, 1000), 9(1000, 1002), 4(1002, 1004), 2(1004, 1011), 3(1011, 1021), 7(1021, 1028), 8(1028, 1038), 6(1038, 1045), 75(1045, 1075) | 8/16 | 1075 |
| 2 | 74(0, 30), 12(30, 772), 26(772, 776), 29(776, 783), 28(783, 788), 27(788, 793), 25(793, 800), 24(800, 805), 14(805, 890), 13(890, 989), 23(989, 996), 18(996, 1004), 15(1004, 1067), 22(1067, 1074), 19(1074, 1082), 21(1082, 1088), 20(1088, 1094), 17(1094, 1101), 16(1101, 1136), 75(1136, 1166) | 13/26 | 1066 |
| 3 | 74(0, 30), 30(30, 856), 45(856, 866), 44(866, 874), 43(874, 882), 41(882, 886), 32(886, 971), 42(971, 981), 31(981, 1080), 40(1080, 1088), 39(1088, 1094), 33(1094, 1100), 34(1100, 1106), 36(1106, 1112), 35(1112, 1118), 38(1118, 1124), 37(1124, 1130), 75(1130, 1160) | 7/14 | 1160 |
| 4 | 74(0, 30), 46(30, 411), 47(411, 673), 63(673, 680), 62(680, 685), 61(685, 690), 49(690, 740), 48(740, 788), 60(788, 795), 59(795, 800), 58(800, 805), 51(805, 814), 57(814, 819), 50(819, 828), 56(828, 833), 55(833, 838), 54(838, 843), 52(843, 992), 53(992, 1139), 75(1139, 1169) | 6/12 | 1169 |
| 5 | 74(0, 30), 64(30, 411), 65(411, 673), 73(673, 680), 66(680, 685), 72(685, 690), 71(690, 695), 68(695, 700), 67(700, 705), 69(705, 710), 70(710, 715), 76(715, 1075) | 3/6 | 1075 |

**Table 4.** Optimization results of ant colony optimization beat 1150 s scheme.

| No. | Time Details for Task Assignment | Tool Switching Times/Time (s) | Accumulated Time |
|---|---|---|---|
| 1 | 74(0, 30), 1(30, 987), 3(987, 997), 8(997, 1005), 5(1005, 1009), 2(1009, 1016), 7(1016, 1021), 6(1021, 1026), 9(1026, 1030), 11(1030, 1037), 10(1037, 1041), 4(1041, 1043), 75(1043, 1073) | 7/14 | 1073 |
| 2 | 74(0, 30), 12(30, 772), 13(772, 871), 25(871, 878), 15(878, 941), 22(941, 948), 27(948, 955), 28(955, 960), 18(960, 968), 26(968, 972), 21(972, 980), 20(980, 986), 17(986, 993), 24(993, 1000), 16(1000, 1035), 29(1035, 1042), 19(1042, 1050), 23(1050, 1057), 75(1057, 1087) | 15/30 | 1087 |
| 3 | 74(0, 30), 46(30, 411), 57(411, 418), 59(418, 423), 55(423, 428), 56(428, 433), 48(433, 483), 47(483, 745), 63(745, 752), 50(752, 761), 49(761, 811), 53(811, 960), 62(960, 967), 51(967, 976), 58(976, 981), 61(981, 986), 54(986, 991), 60(991, 996), 75(996, 1026) | 8/16 | 1026 |
| 4 | 74(0, 30), 65(30, 292), 64(292, 673), 67(673, 680), 66(680, 685), 73(685, 690), 70(690, 695), 71(695, 700), 72(700, 705), 68(705, 710), 69(710, 715), 75(715, 745) | 3/6 | 745 |
| 5 | 74(0, 30), 30(30, 856), 31(856, 955), 33(955, 963), 38(963, 969), 37(969, 975), 43(975, 985), 36(985, 993), 44(993, 1003), 39(1003, 1011), 35(1011, 1017), 42(1017, 1027), 45(1027, 1035), 34(1035, 1043), 40(1043, 1049), 41(1049, 1053), 75(1053, 1083) | 10/20 | 1083 |
| 6 | 74(0, 30), 14(30, 115), 75(115, 145) | 1/2 | 145 |
| 7 | 74(0, 30), 52(30, 179), 75(179, 209) | 1/2 | 209 |
| 8 | 74(0, 30), 32(30, 115), 76(115, 475) | 1/2 | 475 |

It can be seen from Tables 3 and 4 and Figures 11 and 12, that the arrangements of task 3 and task 8, task 12 and task 13, and task 30 and task 31 can satisfy the tight constraint relationship. Meanwhile, from Table 4 and Figure 12, the case of the 'or' constraint can also be analyzed. If task 13 is processed at workstation two, the order constraint of task 31

on 46/47/48/49 disappears. Therefore, we can also see that 46/47/48/49 can indeed be scheduled to be processed at the workstation before task 31, and it shows that the solution obtained by this algorithm can meet the 'or' constraint. Therefore, the algorithm can effectively solve TLBP with multiple complex constraints.

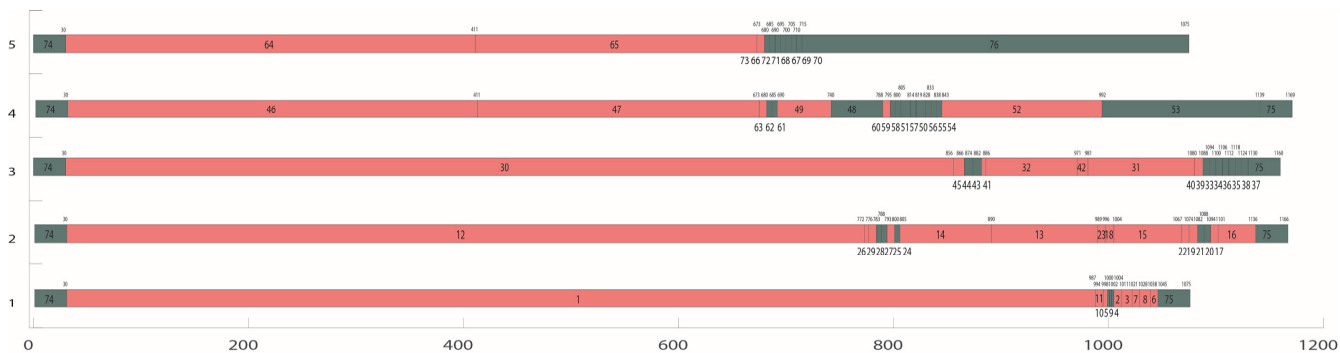

**Figure 11.** Task assignment Gantt chart for cadence time 1170.

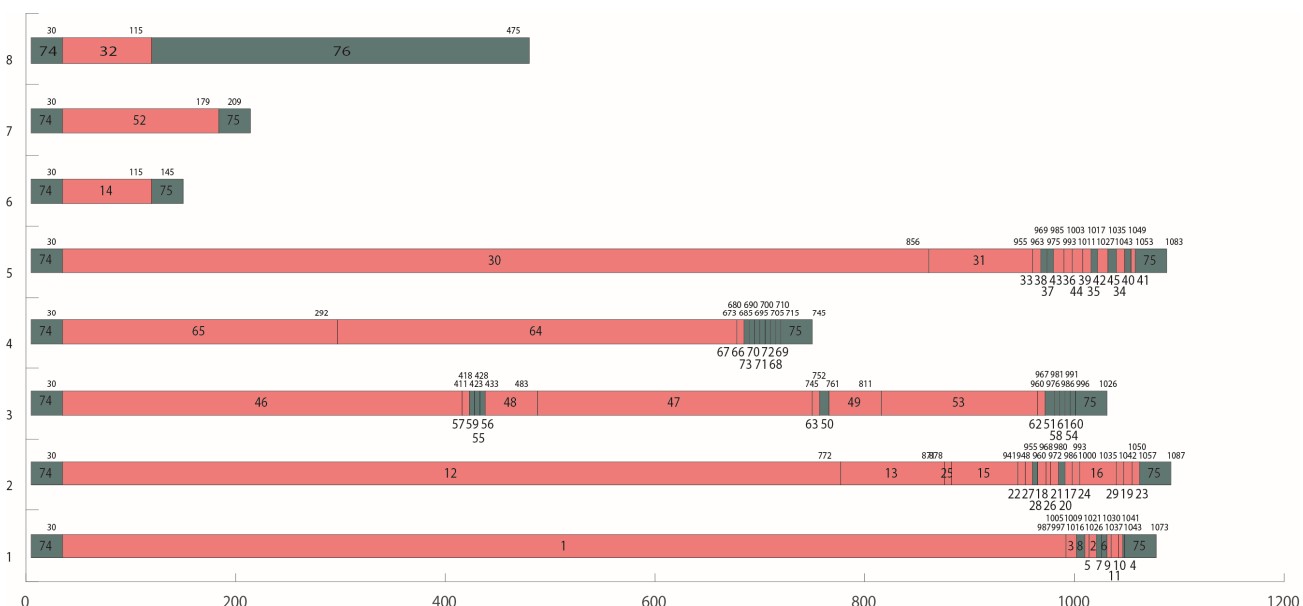

**Figure 12.** Task assignment Gantt chart for cadence time 1150.

Figure 11 shows a Gantt chart of the processing task arrangement obtained from Table 3. In the figure, task 74 is an advanced station preparation task for each station, task 75 is an auxiliary task, such as deburring after each station, and task 76 is a deburring and final inspection task. The Gantt chart was generated using this algorithm. The red task represents the processing of this task; the tool change operation needs to be performed first, and the red task time increases the average tool change time. The blue task indicates that no tool change is required and, therefore, does not include tool change time. The task arrangement in the figure satisfies the task sequence constraints, machining orientation constraints, tool constraints, and auxiliary task constraints, which reflect the practicability of the algorithm in this study.

The Gantt chart in Figure 12 illustrates the allocation scheme in Table 4. From the figure, the imbalance in task assignment among the sites can be seen intuitively. This is the result of a combination of the time, order, tool, and machining direction constraints. Simultaneously, the robustness of the algorithm is verified.

## 7. Discussion

Through the study of a complex case with 73 machining tasks, the algorithm proposed in this study can be used to solve the problem of machining production line balance with multiple complex constraints. And under the premise of satisfying all constraints, the first and last auxiliary tasks proposed in this paper can be inserted into the processing task queue. Due to the existence of auxiliary tasks at the beginning and end of each station in this research question, it is quite different from existing research in constructing task processing queues, and the difference of this processing queue will ultimately directly affect the optimization result after the station is divided. This model considers many practical factors in more machining sites, and it can be said that it is two different branches from the mathematical model of existing research. Therefore, from the perspective of promotion and application, this research has more practical significance. However, in terms of how to improve the processing efficiency of a single station, there are also shortcomings, which is also the direction of further deepening and improvement of this research.

## 8. Conclusions

The balance problem of the machining production line under the combined effect of complex constraints, including tool changing, machining assistance, 'or', and 'tight' constraints, is studied. By designing an improved ant colony optimisation, the multiple screening mechanism of the processing task set satisfies the above complex constraints, the processing task chain is constructed through the ant colony's own pheromone accumulation rules and random search mechanism, and a Gantt chart automatic generation module is designed. Through a detailed analysis of the processing technology case of a certain type of complex box, the positioning benchmark, the evaluation benchmark, and the priority sequence diagram of the processing task were analysed, and reasonable process constraints were established. The influence of random search control parameters on the search performance of the improved ant colony optimisation was studied. The arrangement plan of multiple sets of machining lines was obtained after introducing the case into the algorithm of this study. Through a comparative analysis, it is concluded that the optimisation scheme with a cycle time of 1170 s is the preferred scheme with a balance rate of 95.23%, which has the comprehensive advantages of both cost and efficiency, and it also verifies the performance of the algorithm in this study.

In the future research, optimization objectives such as minimizing the number of tool changes and minimizing the number of machine tools into the mathematical model should be taken into account to study this multi-objective optimization problem. At the same time, in the case of the study, a machining center with an additional fourth axis can also be added to the mathematical model, thereby further increasing the complexity of the problem and the diversification of solutions.

**Author Contributions:** Conceptualisation, J.H.; methodology, J.H.; software, J.H.; validation, J.H., Z.Z. and H.Q.; formal analysis, J.H.; investigation, H.Q.; resources, J.H.; data curation, J.H. and X.X.; writing—original draft preparation, J.H.; writing—review and editing, H.Q.; visualisation, Z.Z.; supervision, Z.Z.; project administration, Z.Z.; funding acquisition, Z.Z. and J.Z. All authors have read and agreed to the published version of the manuscript.

**Funding:** This research was partly supported by the National Natural Science Foundation of China [grant numbers 51205328, 51675450]; the Youth Foundation for Humanities, Social Sciences of Ministry of Education of China [grant number 18YJC630255]; Sichuan Science and Technology Program (Grant No. 2022YFG0245, 2022YFG0241); CRRC's 14th Five-Year Science and Technology Major Special Scientific Research Project(Grant No. 2021CHZ010-3); Special Research Project of Education Department of Zhejiang Province [grant Number Y202146429].

**Institutional Review Board Statement:** Not applicable.

**Informed Consent Statement:** Not applicable.

**Data Availability Statement:** The data presented in this study are available upon request from the corresponding author. The data were not publicly available because of privacy concerns.

**Conflicts of Interest:** The authors declare no conflict of interest.

## Appendix A

Appendix A provides all machining tasks of complex box parts proposed in this study, which are displayed according to the machining planes they are located. All of them are shown in Figures A1–A5.

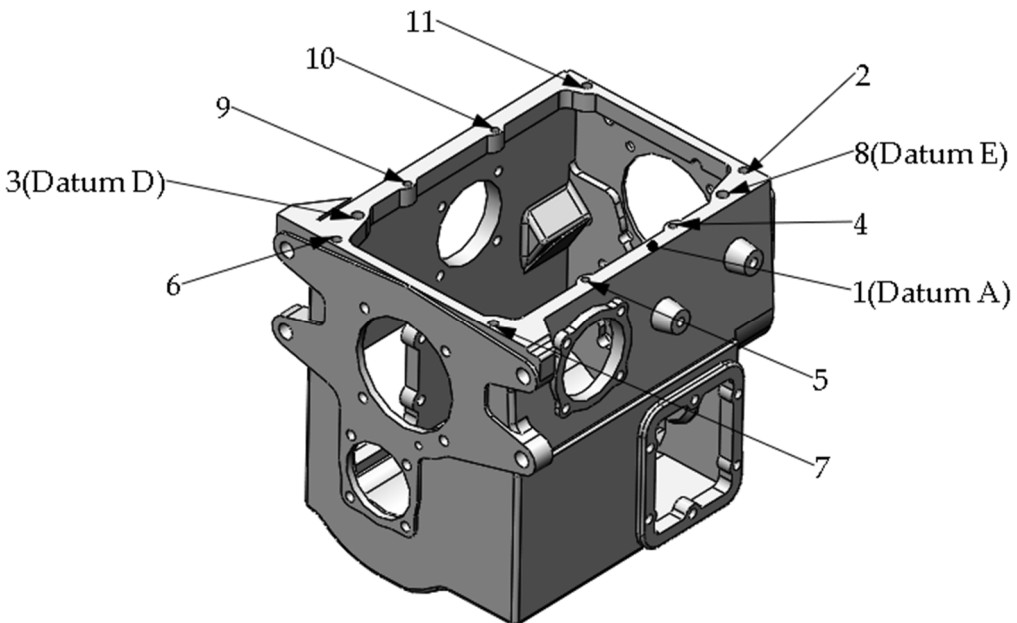

**Figure A1.** Machining tasks on Top surface.

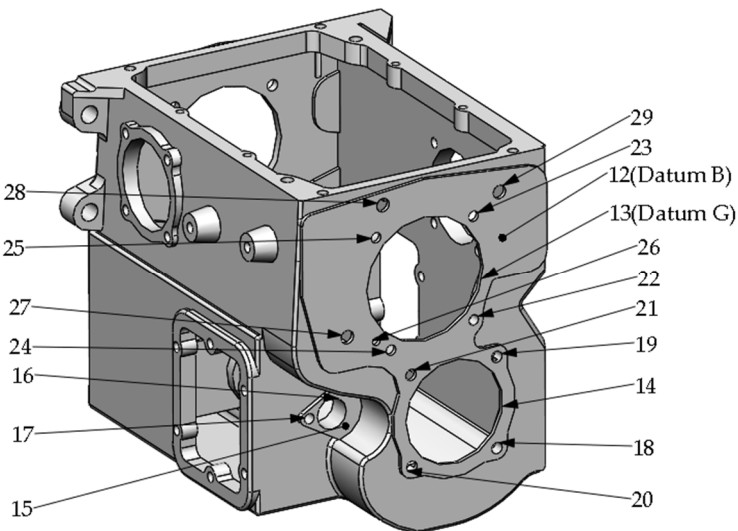

**Figure A2.** Machining tasks on Front surface.

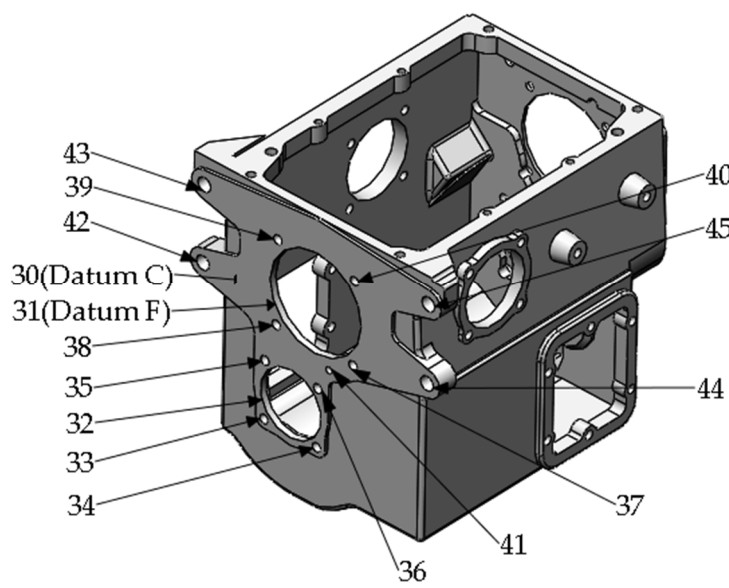

**Figure A3.** Machining tasks on Rear surface.

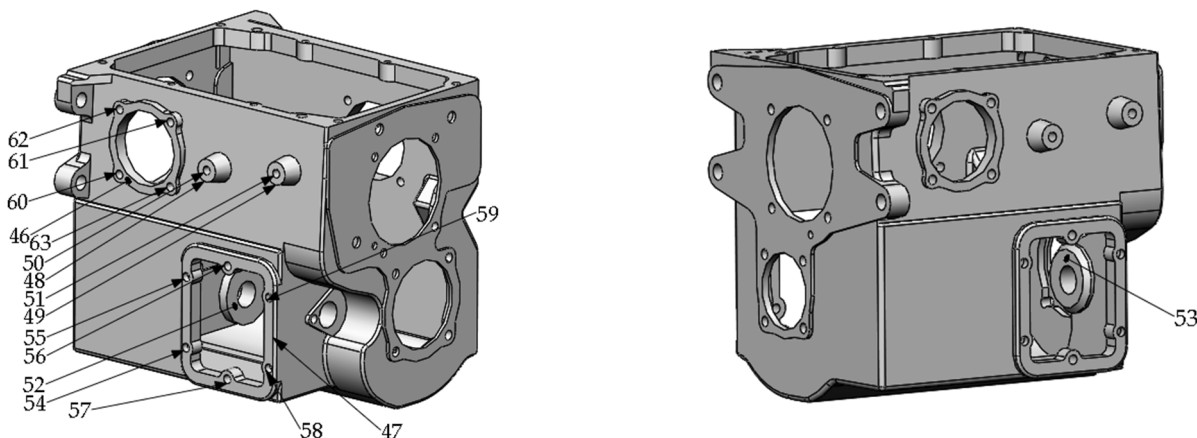

**Figure A4.** Machining tasks on Left surface.

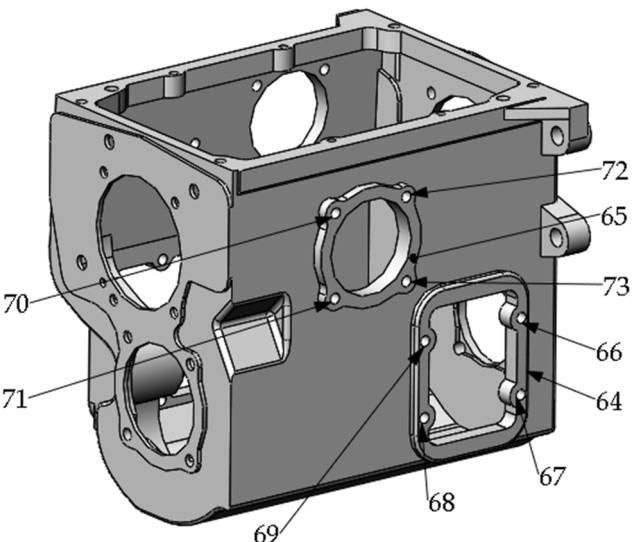

**Figure A5.** Machining tasks on Right surface.

## Appendix B

**Table A1.** Task operation procedure and time.

| Task Number | Processing Content | Processing Times (s) | Plane It Belongs to | Cutting Tool Type | Machining Direction of Parts |
|---|---|---|---|---|---|
| 1 | Datum A | 955 | Datum A | φ30 Milling cutter | Top surface |
| 2 | The hole φ10 on plan A | 5 | Datum A | φ10 combined & drill | Top surface |
| 3 | Hole φ12 (Datum D) | 8 | Datum A | φ12 combined drilling & reaming | Top surface |
| 4 | The hole φ8 on plan A | 2 | Datum A | φ8 combined & drill | Top surface |
| 5 | The hole φ8 on plan A | 2 | Datum A | φ8 combined & drill | Top surface |
| 6 | The hole φ10 on plan A | 5 | Datum A | φ10 combined & drill | Top surface |
| 7 | The hole φ10 on plan A | 5 | Datum A | φ10 combined & drill | Top surface |
| 8 | Hole φ12 (Datum E) | 8 | Datum A | φ12 combined drilling & reaming | Top surface |
| 9 | The hole φ8 on plan A | 2 | Datum A | φ8 combined & drill | Top surface |
| 10 | The hole φ8 on plan A | 2 | Datum A | φ8 combined & drill | Top surface |
| 11 | The hole φ10 on plan A | 5 | Datum A | φ10 combined & drill | Top surface |
| 12 | Plane B (Datum B) | 740 | Datum B | φ50 Milling cutter | Front surface |
| 13 | Hole G (Datum G) | 97 | Datum B | φ50 Milling cutter φ120 boring cutter | Front surface |
| 14 | The hole φ100 on plan B | 83 | Datum B | φ50 Milling cutter φ100 boring cutter | Front surface |
| 15 | Alcove | 61 | Alcove | φ50 Milling cutter | Front surface |
| 16 | Hole φ30 on the alcove | 33 | Alcove | φ30 Milling cutter | Front surface |
| 17 | Hole φ10 on the alcove | 5 | Alcove | φ10 spiral drill | Front surface |
| 18 | The hole φ11 on plan B | 6 | Datum B | φ11 combined & drill | Front surface |
| 19 | The hole φ11 on plan B | 6 | Datum B | φ11 combined & drill | Front surface |
| 20 | The hole φ11 on plan B | 6 | Datum B | φ11 combined & drill | Front surface |
| 21 | The hole φ11 on plan B | 6 | Datum B | φ11 combined & drill | Front surface |
| 22 | The hole φ10 on plan B | 5 | Datum B | φ10 combined & drill | Front surface |
| 23 | The hole φ10 on plan B | 5 | Datum B | φ10 combined & drill | Front surface |
| 24 | The hole φ10 on plan B | 5 | Datum B | φ10 combined & drill | Front surface |
| 25 | The hole φ10 on plan B | 5 | Datum B | φ10 combined & drill | Front surface |
| 26 | The hole φ8 on plan B | 2 | Datum B | φ8 combined & drill | Front surface |
| 27 | The hole φ13 on plan B | 5 | Datum B | φ13 combined & drill | Front surface |
| 28 | The hole φ13 on plan B | 5 | Datum B | φ13 combined & drill | Front surface |
| 29 | The hole φ13 on plan B | 5 | Datum B | φ13 combined & drill | Front surface |
| 30 | Plane C (Datum C) | 824 | Datum C | φ50 Milling cutter | Rear surface |
| 31 | Hole F (Datum F) | 97 | Datum C | φ50 Milling cutter φ120 boring cutter | Rear surface |
| 32 | The hole φ100 on plan C | 83 | Datum C | φ50 Milling cutter φ100 boring cutter | Rear surface |
| 33 | The hole φ11 on plan C | 6 | Datum C | φ11 combined & drill | Rear surface |
| 34 | The hole φ11 on plan C | 6 | Datum C | φ11 combined & drill | Rear surface |
| 35 | The hole φ11 on plan C | 6 | Datum C | φ11 combined & drill | Rear surface |
| 36 | The hole φ11 on plan C | 6 | Datum C | φ11 combined & drill | Rear surface |
| 37 | The hole φ11 on plan C | 6 | Datum C | φ11 combined & drill | Rear surface |
| 38 | The hole φ11 on plan C | 6 | Datum C | φ11 combined & drill | Rear surface |
| 39 | The hole φ11 on plan C | 6 | Datum C | φ11 combined & drill | Rear surface |
| 40 | The hole φ11 on plan C | 6 | Datum C | φ11 combined & drill | Rear surface |
| 41 | The hole φ8 on plan C | 2 | Datum C | φ8 combined & drill | Rear surface |
| 42 | The hole φ17 on plan C | 8 | Datum C | φ17 combined & drill | Rear surface |
| 43 | The hole φ17 on plan C | 8 | Datum C | φ17 combined & drill | Rear surface |
| 44 | The hole φ17 on plan C | 8 | Datum C | φ17 combined & drill | Rear surface |
| 45 | The hole φ17 on plan C | 8 | Datum C | φ17 combined & drill | Rear surface |
| 46 | rectangular window at right side of the part | 379 | rectangular window (right side of the part) | φ50 Milling cutter | Left side |
| 47 | round window at right side of the part | 260 | round window (right side of the part) | φ20 Milling cutter | Left side |
| 48 | Lug boss 1 at right side of the part | 48 | Lug boss (right side of the part) | φ25 Milling cutter | Left side |
| 49 | Lug boss 2 at right side of the part | 48 | Lug boss (right side of the part) | φ25 Milling cutter | Left side |
| 50 | Hole φ10 on Lug boss 1 | 9 | Lug boss (right side of the part) | φ10 combined & drill | Left side |
| 51 | Hole φ10 on Lug boss 2 | 9 | Lug boss (right side of the part) | φ10 combined & drill | Left side |

**Table A1.** *Cont.*

| Task Number | Processing Content | Processing Times (s) | Plane It Belongs to | Cutting Tool Type | Machining Direction of Parts |
|---|---|---|---|---|---|
| 52 | The right wall of rectangular window | 147 | rectangular window (right side of the part) | φ30 Long shank milling cutter | Left side |
| 53 | The left wall of rectangular window | 147 | rectangular window (right side of the part) | φ30 Long shank milling cutter | Left side |
| 54 | Hole φ10 on rectangular window | 5 | rectangular window (right side of the part) | φ10 combined & drill | Left side |
| 55 | Hole φ10 on rectangular window | 5 | rectangular window (right side of the part) | φ10 combined & drill | Left side |
| 56 | Hole φ10 on rectangular window | 5 | rectangular window (right side of the part) | φ10 combined & drill | Left side |
| 57 | Hole φ10 on rectangular window | 5 | rectangular window (right side of the part) | φ10 combined & drill | Left side |
| 58 | Hole φ10 on rectangular window | 5 | rectangular window (right side of the part) | φ10 combined & drill | Left side |
| 59 | Hole φ10 on rectangular window | 5 | rectangular window (right side of the part) | φ10 combined & drill | Left side |
| 60 | Hole φ10 on round window | 5 | round window (right side of the part) | φ10 combined & drill | Left side |
| 61 | Hole φ10 on round window | 5 | round window (right side of the part) | φ10 combined & drill | Left side |
| 62 | Hole φ10 on round window | 5 | round window (right side of the part) | φ10 combined & drill | Left side |
| 63 | Hole φ10 on round window | 5 | round window (right side of the part) | φ10 combined & drill | Left side |
| 64 | rectangular window at left side of the part | 379 | rectangular window (left side of the part) | φ50 Milling cutter | Right side |
| 65 | round window at left side of the part | 260 | round window (left side of the part) | φ20 Milling cutter | Right side |
| 66 | Hole φ10 on rectangular window | 5 | rectangular window (left side of the part) | φ10 combined & drill | Right side |
| 67 | Hole φ10 on rectangular window | 5 | rectangular window (left side of the part) | φ10 combined & drill | Right side |
| 68 | Hole φ10 on rectangular window | 5 | rectangular window (left side of the part) | φ10 combined & drill | Right side |
| 69 | Hole φ10 on rectangular window | 5 | rectangular window (left side of the part) | φ10 combined & drill | Right side |
| 70 | Hole φ10 on round window | 5 | round window (left side of the part) | φ10 combined & drill | Right side |
| 71 | Hole φ10 on round window | 5 | round window (left side of the part) | φ10 combined & drill | Right side |
| 72 | Hole φ10 on round window | 5 | round window (left side of the part) | φ10 combined & drill | Right side |
| 73 | Hole φ10 on round window | 5 | round window (left side of the part) | φ10 combined & drill | Right side |
| 74 | The preparation for the head of each station | 30 | NA | Inspection of cleaning appearance | All |
| 75 | chip removing at the end of each station | 30 | NA | chip removing tool | All |
| 76 | inspection by Position gage | 360 | NA | Position gauge | All |

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
