# Peer review of "Enhanced Hybrid Ant Colony Optimization for Machining Line Balancing Problem with Compound and Complex Constraints"

_applsci, doi:10.3390/app12094200_

Round 1

Reviewer 1 Report

This paper is about enhanced hybrid Ant Colony Optimization for machining line balancing problem with compound and complex constraints. I have the following comments:

  1. As mentioned by Denkena et al.[4], ---> references must be preceded by space
  2. The paper needs extensive English editing. For example, this sentence is very long (8 lines) “As mentioned by Denkena et al.[4], the detailed steps of the process planning process include receiving design drawings and analysing design details, selecting raw materials of suitable shapes, selecting process technology, deciding process sequence and flow, pre- paring processing plans (equipment requirements, flow sequence of products in the equipment, cutters and tools, processing step design, processing parameter selection, measuring tools and tooling design, tool path planning, process cost, and time analysis), and, finally, obtaining an executable CNC program, process cards, operating procedures, and other process documents”
  3. The motivation of the work is not well presented. The Introduction needs to be modified to add the motivation of the work
  4. [25,26] ---> separate the two references by a space
  5. After related work and after you state the limitations of the related work, add a statement to clearly identify the contributions of the system. For example, the main contributions are ….
  6. What is the objective /cost function that needs to be optimized?
  7. Explain more about the type of the optimization problem. Is it general nonlinear? Complex optimization, etc?
  8. What are the limitations of the proposed work?
  9. What is the complexity of the proposed model?
  10. Add a section to talk about future work

Reviewer 2 Report

This paper discusses the balance problem of machining production lines under the combined effect of complex constraints, including tool changing, machining assistance, 'or', and 'tight' constraints. The proposed approach is quite interesting to solve some real scheduling problems. I want to recommend publication of the paper after the authors have addressed the concerns I have stated below.

1.
What is the meaning of TLBP. I cannot understand what is the meaning of "T".

2.
I wonder that the Gantt diagram in Figure 4 does not consider the sequence constraints described in Figure 2.

3.
In line 278, I cannot find the definition of "t_i" and "|e^{k-m}|".

4.
In lines 414-417, the authors explain the evaluation criteria of the target problem. But I cannot understand why A, D, E benchmarks are most important in this problem. Please describe the details.

5.
In this case study, ACO, ACOpro1, and ACOpro2 are evaluated, and ACOpro2 is used finally. However, I think the general superiority of ACOpro2 cannot be explained only from this case study. So, why did the authors compare these three algorithms?

6. 
I think the most interesting originality of this paper is considering "or" and "tight" constraints. However, in Chapter 6, there is no demonstration of considering these constraints. I think the authors should demonstrate the originality clearly in the case study.

Reviewer 3 Report

The paper is of good structure, proper length and interesting, important and up-to-date topic. Although the presented approach is based on well-known and widely used mixed-integer programming the paper introduces a novel solution because of simultaneous implementation heuristics task set filtering mechanism and ant colony optimization method. Therefore the paper is worth of publishing and can be interesting of many researchers. However, I have some remarks and suggestions that can help to improve the paper and should be taken into account before its publication:

  1. Introduction – In my opinion the introduction is not directly connected with the title and rest of the paper. In fact you discuss the problem of computer-process planning while the paper is focused on the problem of machining line balancing problem. In my opinion in the introduction of the paper you should emphasize the importance of the machining line balancing problem in modern manufacturing systems and necessity of providing further research in this area.
  2. You should define clearly the goal of your paper.
  3. Literature analysis. In my opinion the provided literature analysis is very cursory. It is very easy to find many papers directly connected with the topic of the paper that not have been mentioned (see e.g. paper with DOIs: 10.1007/978-3-030-91059-4_9; 10.1109/Dynamics50954.2020.9306146; 10.1007/978-3-319-24834-9_31; 10.1080/00207543.2019.1593549, etc.). Therefore you should make deeper and more detailed literature review (especially that you write – line 164 – that “this paper also considered all constraints reported in existing literature”).
  4. The diagram presented in Fig 2 is not very clear (it should be explained more clearly). In fact the diagram suggests that it is not necessary to make all machining operations to make the product – i.e. the graph suggests that the part can be machined by realizing operations 1, 2, 5, 6, or 1, 2, 5, 8 or 1, 3, 4, 8 or 1, 3, 7. Is it really true?
  5. The Gantt diagram presented in Fig. 4 is not very accurate. In fact it suggests that operations with times t1, t2, t4, t5 and t6 are realized in the same time. Moreover you assume that preparation time (set-up time) is equal at each machine for each operation is the same. It not seems to be possible in real industrial conditions.

Reviewer 4 Report

Dear Editor and authors, in reviewed manuscript authors presents their work on Enhanced Hybrid Ant Colony Optimization for Machining Line Balancing Problem with Compound and Complex Con- straitns. I have carefully reviewed the manuscript and listed some comments and suggestions, which can/must be improved.

My opinion on Introduction section, appropriate research question statement. Appropriate and clearly defined related work section. In addition (section 2.2) seme new references can be added as reference: https://doi.org/10.2507/ijsimm20-4-578.

In Section 3, I would suggest that you add some text between section 3 and 3.1. If not just start writing the description of the evaluated bracket.

Table 1, if the t1, t2, etc. are variables, II think that they are, use italic text! All variables (like pz, aij, etc., especially form line 290 to 310) in the manuscript must be written in italic, reconsider in entire manuscript.

In my opinion, if you refer to the equation the correct citation in the text must be added and used.

In section 4, in my opinion you should proposed other algorithms as a test benchmark too. If you are testing efficiency just with your algorithm it is hard to know, how better, more robust, reliable your proposed algorithm is. In my opinion this should be extended in the manuscript. You have made to partial answer to this question in section 6, but it must be extended according to the existing literature.

On almost all of your figures some links appear, showing path form where the figure was added (form PC to text). In my opinion you can delete this links.

Please be more specific when using decimal dot (.) and I suggest that you are using decimal comma (,) where the number is higher than three digits value.

If possible, reconsider Figure 11, two lines structure, it would be much more readable and clearly visible. On fi. 12, you have use to much zoom in on x, y values but not too much on Gantt chart values. Reconsider that.

I miss the discussion section in my opinion in discussion section, authors should answer following questions:

  • How do these results relate to the original question or objectives outlined in the Introduction section?
  • Do the data support your initial research question?
  • Are your results consistent with what other investigators have made?
  • Some general discussion was made in relation to weaknesses and discrepancies.

In section Conclusions, I’m missing what further research would be necessary to answer the questions raised by your results?

Round 2

Reviewer 1 Report

The authors have addressed my comments

Reviewer 2 Report

The manuscript has been much improved and is in nice condition now. I think this manuscript will be acceptable.

Reviewer 3 Report

Please include my remarks and suggestions provided in my previous review (unfortunately your improvements are very cursory and they are not suitable when take into account). Please that I asked:

  1. To rebuild the Introduction section. Introduction of the paper you should emphasize the importance of the machining line balancing problem in modern manufacturing systems and necessity of providing further research in this area.
  2. You should make deeper and more detailed literature review. I have indicated some papers directly connected with the topic of your paper that should been taken into account. Of course other previously papers connected with the topic of your paper are welcomed.
  3. The diagram presented in Fig 2 should be clarified in the body of the text. It is ambiguous and therefore should be explained more clearly.
  4. Following the previous remark also the Gantt diagram presented in Fig. 4 should be explained more clearly (in the body of your paper). It still suggests that operations with times t1, t2, t4, t5 and t6 are realized in the same time. Moreover, you assume that preparation time (set-up time) is equal at each machine for each operation is the same. It not seems to be possible in real industrial conditions.
